# Mechanism of actin-dependent activation of nucleotidyl cyclase toxins from bacterial human pathogens

Alexander Belyy [1], Felipe Merino [1,2], Undine Mechold[3] & Stefan Raunser [1✉]

Bacterial human pathogens secrete initially inactive nucleotidyl cyclases that become potent enzymes by binding to actin inside eukaryotic host cells. The underlying molecular mechanism of this activation is, however, unclear. Here, we report structures of ExoY from *Pseudomonas aeruginosa* and *Vibrio vulnificus* bound to their corresponding activators F-actin and profilin-G-actin. The structures reveal that in contrast to the apo-state, two flexible regions become ordered and interact strongly with actin. The specific stabilization of these regions results in an allosteric stabilization of the nucleotide binding pocket and thereby to an activation of the enzyme. Differences in the sequence and conformation of the actin-binding regions are responsible for the selective binding to either F- or G-actin. Other nucleotidyl cyclase toxins that bind to calmodulin rather than actin undergo a similar disordered-to-ordered transition during activation, suggesting that the allosteric activation-by-stabilization mechanism of ExoY is conserved in these enzymes, albeit the different activator.

[1] Department of Structural Biochemistry, Max Planck Institute of Molecular Physiology, Otto-Hahn-Str. 11, 44227 Dortmund, Germany. [2] Department of Protein Evolution, Max Planck Institute for Developmental Biology, Max-Planck-Ring 5, 72076 Tübingen, Germany. [3] Unité de Biochimie des Interactions Macromoléculaires, Département de Biologie Structurale et Chimie, Institut Pasteur, CNRS UMR 3528 Paris, France. ✉email: stefan.raunser@mpi-dortmund.mpg.de

*P*seudomonas aeruginosa and *Vibrio vulnificus* are human pathogens that cause nosocomial infections and seafood-related illnesses, respectively, with severity ranging from benign and local to systemic and life-threatening[1,2]. A major virulence determinant of *P. aeruginosa* is the type 3 secretion system (T3SS), a cell wall associated nanomachine which transports a panel of effectors into eukaryotic target cells[3,4]. One of such T3SS effectors is the nucleotidyl cyclase toxin exoenzyme Y (PaExoY)[5]. It produces a supraphysiologic amount of 3′,5′-cyclic guanosine monophosphate (cGMP) and 3′,5′-cyclic adenosine monophosphate (cAMP) and thereby disorganizes cell signaling[6]. At high concentrations, this results in the death of the intoxicated cultured cells[7,8]. At lower PaExoY concentrations, which are more likely to occur during chronical infection, the activity of PaExoY allows *P. aeruginosa* to hijack host immune response by a suppression of TAK1 and a decreased production of interleukin 1[9,10]. A recent report has established a link between the presence of PaExoY in clinical isolates and end-organ dysfunction in critically ill patients[11].

*V. vulnificus* and its close relative *V. nigripulchritudo* also possess ExoY-like toxins (VvExoY and VnExoY, respectively), which act as adenylate cyclases[12]. VvExoY is essential for virulence in mice and is responsible for severe tissue damage[13]. In contrast to PaExoY, VvExoY and VnExoY are not delivered by T3SS but as modules of MARTX (Multifunctional-Autoprocessing Repeats-in-Toxin). These large toxins combine toxic effector modules with their delivery apparatus in a single polypeptide chain[14].

Both PaExoY and VvExoY are catalytically inactive inside bacteria. However, once delivered into the target eukaryotic cell, the effectors bind to actin, resulting in more than 10,000-fold increase in nucleotidyl cyclase activity[15]. A similar activation potency was observed in other members of nucleotidyl cyclase family, such as the edema factor of *Bacillus anthracis* and the adenylate cyclase domain of CyaA of *Bordetella pertussis*, which are activated by calmodulin. A series of structural studies revealed that interaction with calmodulin promotes a disordered-to-ordered transition in the catalytic center that is necessary for the efficient catalysis[16–18]. However, it is not known if activation of nucleotidyl cyclases by actin follows the same molecular mechanism.

Here we report cryo-EM structures of PaExoY and VvExoY in complex with their activators F-actin and G-actin-profilin. Our structural data together with molecular dynamics simulations and enzymatic assays reveal that specific interactions between two independent regions of the toxins with F-, or G-actin lead to stabilization of the catalytic center of the toxins and high enzymatic activity. Thus, this mechanism appears to be general for both bacterial cyclase toxin families.

## Results

**P. aeruginosa ExoY-F-actin complex**. To decipher the activation mechanism of PaExoY, we reconstituted the complex of the toxin with the substrate analog 3′-deoxyguanosine triphosphate (3′-dGTP) and F-actin in vitro. We also added phalloidin, a fungal toxin that stabilizes F-actin and thus decreases the concentration of non-polymerized actin in the sample[19,20]; both effects are beneficial for single particle cryo-EM. Phalloidin does not change the affinity of PaExoY to F-actin[21]. We then applied single particle cryo-EM to determine the structure of the complex at an average resolution of 3.2 Å which allowed us to build an almost full atomic model of the complex (Fig. 1a, Supplementary Fig. 1, Supplementary Table 1). PaExoY binds to F-actin in a 1:1 stoichiometry, which is consistent with our previous biochemical experiments[15]. The absence of density between consecutive PaExoY monomers along the filament shows that there is no interaction between them.

The overall structure of PaExoY bound to F-actin (Fig. 1b and c) is similar to the structure of PaExoY in the apo state[22] (Supplementary Fig. 2a) with a 2.7 Å root-mean-square deviation (RMSD) between all common atoms. PaExoY is composed of a smaller actin-binding domain (ABD) that is connected via a hydrophobic core (Supplementary Fig. 2b) to a larger catalytic domain with the nucleotide binding pocket (NBP) in its center. In contrast to the crystal structure of the apo state, which is based on an enzyme that was treated by limited in situ proteolysis to remove flexible regions, the NBP and ABD are resolved in the PaExoY-F-actin cryo-EM map (Supplementary Fig. 2a). The ABD adapts to the surface of the filament and forms two extensive contacts with subdomains I and III of actin (Fig. 1d, Supplementary Fig. 2a and c).

The central contact region (amino acids Ile-222–Glu-258) interacts with the nucleotide-binding region and subdomain I of actin. At the end of this interface sits a loop with three aromatic residues that create a scaffold for the formation of a salt bridge between Asp-247 of PaExoY and Arg-95 of F-actin (Supplementary Fig. 2c). We performed a thorough mutational analysis of this region and either depleted the aromatic residues or the salt bridge and tested the binding affinity and enzymatic activity in vitro (Supplementary Fig. 2d, e, g). Interestingly, these mutations did not affect the affinity of the toxin to F-actin but tremendously impaired its enzymatic activity. We then employed the commonly used *Saccharomyces cerevisiae* model to assess toxicity of the PaExoY variants[23]. Our previous studies demonstrated high sensitivity of this model as activation of endogenously expressed PaExoY by yeast actin leads to rapid cell death[21]. In agreement with the in vitro studies, the mutations in PaExoY decreased toxicity in yeast (Supplementary Fig. 2f). Thus, while this central contact region does not strongly contribute to the affinity of the complex, it plays an essential role in toxin activation by detecting the presence of the activator. Therefore, we decided to term this region 'sensor'.

The other PaExoY-F-actin interface with a buried surface area of 750 Å$^2$ is formed by the C-terminal region of PaExoY (amino acids Lys-347–Val-378). It is composed of a loop and a helix, interacting with a hydrophobic groove on the surface of subdomain I of actin (Fig. 1d). Interestingly, Phe-374 and Val-378 at the tip of the helix extend over to the hydrophobic interface between two neighboring actin subunits (Fig. 1d). Indeed, deletions of the 5 C-terminal residues in PaExoY completely abolished binding to F-actin and activity (Supplementary Fig. 2d–g), suggesting that unlike the sensor, this region, which we term "anchor", largely determines the 1 µM affinity of the toxin to F-actin. Since this pocket exists only in filamentous actin, Phe-374 and Val-378 are likely key residues that provide specificity for F-actin.

**Mechanism of PaExoY activation**. The NBP is conserved between members of the nucleotidyl cyclase toxins[5,22,24]. Our structure of PaExoY in complex with F-actin reveals now the atomic details of the active center of an actin-activated nucleotidyl cyclase. In the active center of PaExoY, Arg-63, Lys-81, and Lys-88 interact with phosphate groups of 3′-dGTP and orient the nucleotide for hydrolysis (Fig. 2a). Phe-83, Glu-258, Ser-292, Asn-297, and Pro-298 create a pocket that harbors the base of the nucleotide (Fig. 2a). Finally, a third group of residues, including Asp-212, Asp-214, and His-291 are localized in the proximity of the ribose of 3′-dGTP. Since His-291 is localized ~4 Å away from the ribose, it probably acts as a nucleophilic base for deprotonation of 3′OH group of GTP (Fig. 2a). There is an ongoing debate whether one or two Mg$^{2+}$ ions are required for the efficient catalysis of bacterial nucleotidyl cyclases[16,25]. In our structure, we see only a weak density at the position, where Mg$^{2+}$

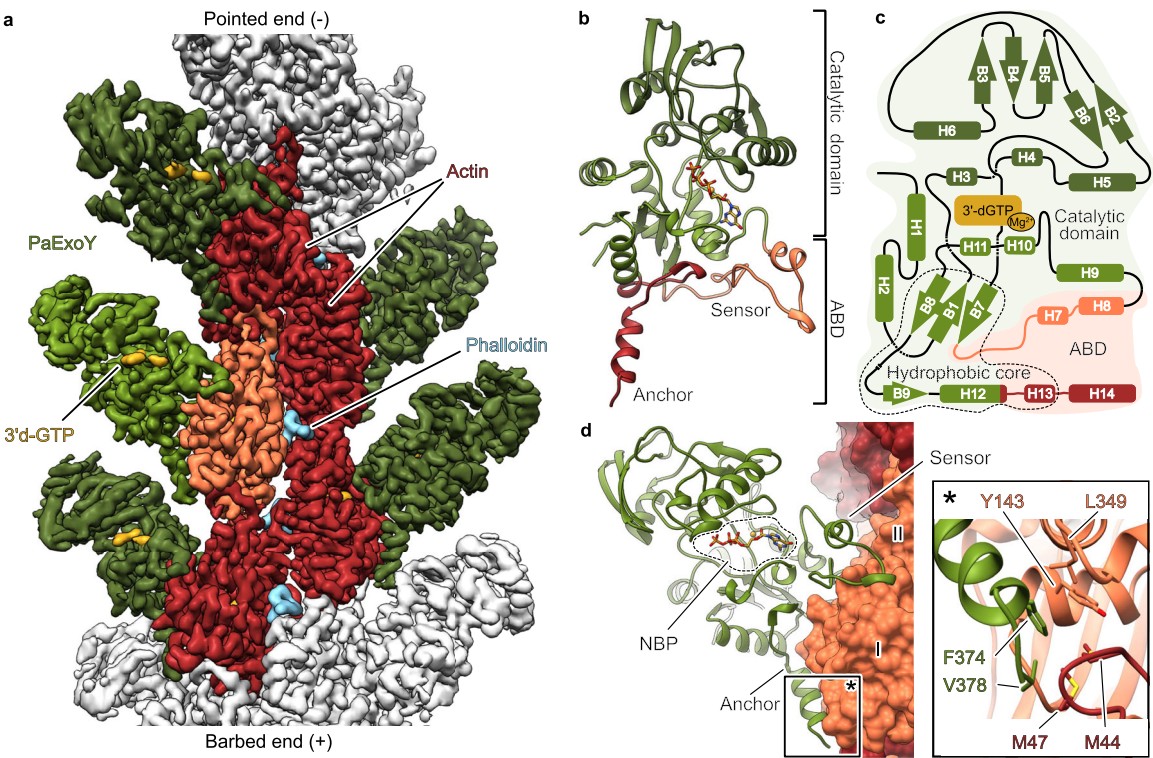

**Fig. 1 Cryo-EM structure of PaExoY in complex with F-actin. a** The postprocessed map of the PaExoY-F-actin complex. The central actin subunit is colored in orange. Its surrounding 4 neighbors are shown in dark red. PaExoY is in green. **b** Domain organization of PaExoY. **c** Schematic representation of the secondary structure elements of PaExoY. **d** Overview of the PaExoY-F-actin interface. Actin subdomains are shown in surface representation and are marked by roman numerals. ABD actin-binding domain, NBP nucleotide-binding pocket.

presumably coordinates the β and γ phosphates of the nucleotide, but we observe a clear density for a metal ion, likely $Mg^{2+}$, that is coordinated by Asp-212, Asp-214, and His-291 (Supplementary Fig. 2h and i).

Due to flexibility, the lower part of the NBP has been absent in the crystal structure of PaExoY in the apo state[22] (Supplementary Fig. 2a). Stabilization of the pocket induced by the interaction of PaExoY with F-actin would therefore explain the strong activation effect of F-actin on the toxin.

To further explore the mechanism of activation of PaExoY, we performed molecular dynamics (MD) simulations of the toxin with and without F-actin, starting from its actin-bound conformation. In the absence of F-actin, PaExoY quickly drifts away from its starting conformation, with particularly strong fluctuations in the ABD and in parts of the NBP (Fig. 2b, Supplementary Fig. 3a and b, Supplementary Movie 1). In strong contrast, in simulations of the PaExoY-F-actin complex, the toxin reproducibly maintains its activated conformation, even when no substrate is present at its active site (Fig. 2b, Supplementary Fig. 3a and b, Supplementary Movie 1). Moreover, key interactions, including Arg-95-Asp-247 and Asp-25-Lys-347/Arg364 (in F-actin and PaExoY, respectively), are consistently maintained in all simulations of the complex (Supplementary Fig. 4). Overall, the calculations suggest that a part of PaExoY undergoes a disordered-to-ordered transition upon an induced fit binding to F-actin, which results in the activation of the enzyme.

To decipher how the activation is communicated between the peripheral ABD and the central NBP, we applied dynamic network analysis to the MD simulation results[26]. With this method, amino acids are represented as nodes in a network. Only amino acids with stable interactions during the simulation are connected, and their connection strength (the edge weight) is

proportional to the correlation between the motion of the two amino acids in the simulations.

The dynamic network analysis revealed that structural changes in the ABD are transmitted to the center of PaExoY via two distinct pathways. While changes at the anchor and at the adjacent part of the sensor are transferred to the NBP via the hydrophobic core, changes at the peripheral part of the sensor are directly transmitted to the NBP to helix 10 (Fig. 2c). Arg-233 and Asp-293 are central residues of the latter pathway, forming a salt bridge between the sensor and the NBP (Supplementary Fig. 3c). Single amino acid substitutions of these residues fully abolished toxin activation but not F-actin binding (Supplementary Fig. 3d, f and g), corroborating their direct involvement in the communication of the activation signal from the periphery to the center of the toxin. The same is true for the hydrophobic core. A point mutation in its core abolished toxin activation (Supplementary Fig. 3d, f and g), indicating that proper packing of this area is key for allosteric signal transduction.

**_V. vulnificus_ ExoY-G-actin-profilin complex.** In contrast to PaExoY, which is activated by F-actin, its homologs from _Vibrio nigripulchritudo_ (VnExoY)[12] and the human pathogen _Vibrio vulnificus_ (VvExoY) (Supplementary Fig. 5) are activated by G-actin, indicating a high level of specificity of ExoYs for their activator. To decipher the molecular basis for this specificity, we set out to solve structures of VvExoY and VnExoY. We started with VnExoY in complex with 3′-dATP and G-actin. However, although we tested different constructs, actin isoforms and cryo conditions, strong anisotropy prevented us from obtaining a high-resolution reconstruction (Supplementary Fig. 6). We, therefore, tried the equivalent VvExoY complex instead, but faced

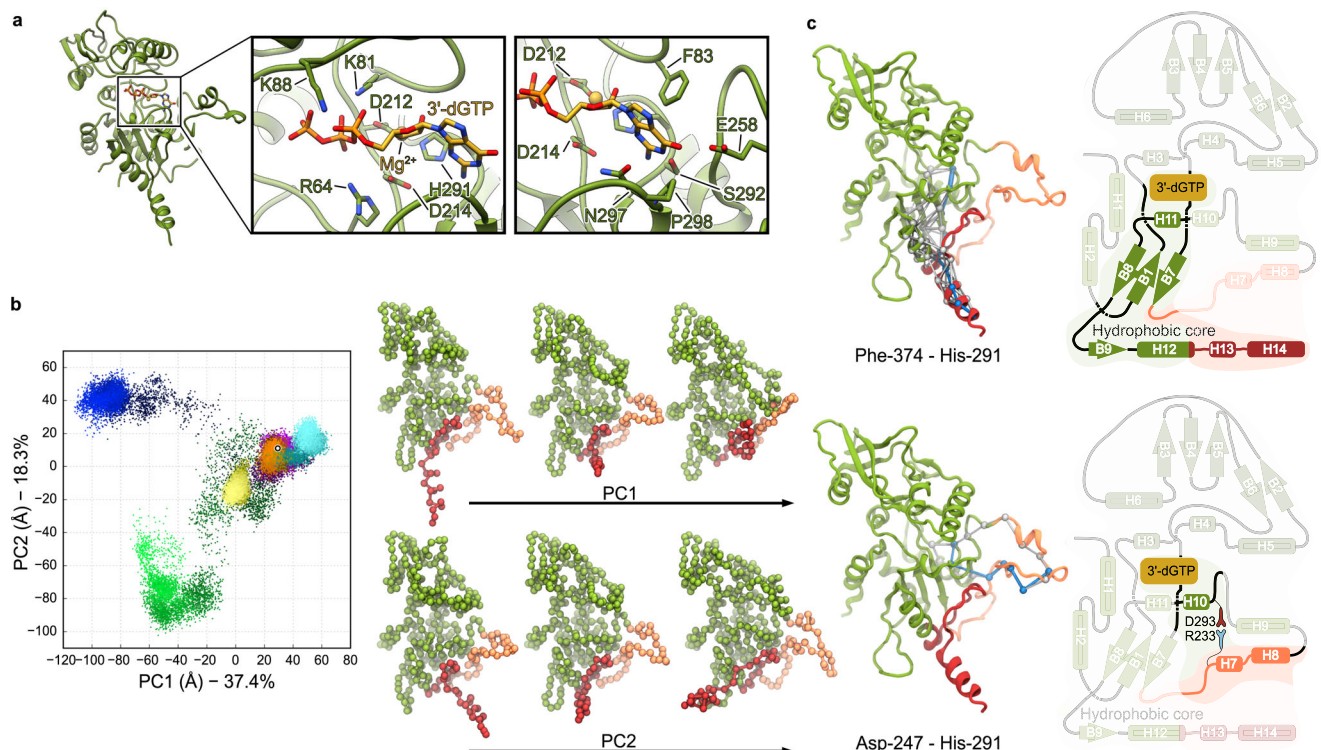

**Fig. 2 Effect of F-actin binding on the structure of PaExoY. a** Atomic model of the PaExoY with focus on the nucleotide-binding pocket (NBP). **b** Principal component (PC) analysis of the different simulations. Each point represents a conformation projected onto the first two principal components. For each trajectory, time progresses as the color goes from dark to bright. Blue and green correspond to free PaExoY, while yellow, cyan, orange, and purple correspond to the F-actin-PaExoY complex. The white dot represents the starting conformation for all simulations. The structures on the right from the plot show the nature of the variation along each component. **c** Collection of the paths (white) with the shortest possible path (light blue) connecting His-291 at the active site of PaExoY with different regions interacting with actin.

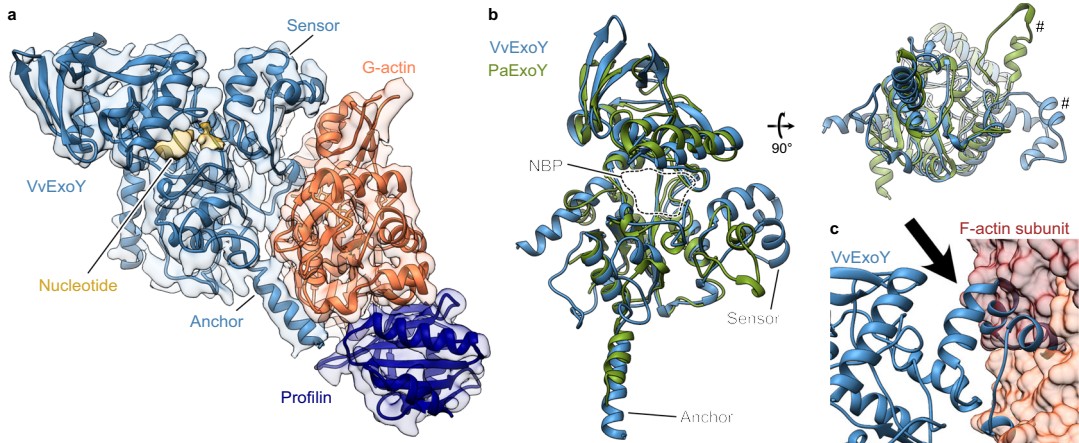

**Fig. 3 Cryo-EM structure of the VvExoY-G-actin-profilin complex. a** An atomic model of the complex fit in the cryo-EM density that was combined from two maps (Supplementary Fig. 7c). VvExoY is in light blue, G-actin is in orange, profilin is in dark blue. **b** Alignment of atomic models of PaExoY (green) and VvExoY (light blue). Hashtags indicate the major difference in the sensor of the toxins. NBP nucleotide-binding pocket. **c** Docking of the VvExoY structure to F-actin shows a steric clash (arrow) between the central interaction region of VvExoY and the F-actin subunit.

the same problem. Often, additional components of a protein complex change the characteristics of a particle strongly enough that they distribute differently in the ice. We, therefore, decided to add the actin-binding protein profilin to the complex, which does not interfere with the activation of VvExoY (Supplementary Fig. 5). Since most of G-actin in cells is captured by profilin, this represents well the situation in vivo. The addition of profilin improved anisotropy and allowed us to determine the structure of the VvExoY-G-actin-profilin complex at 3.9 Å resolution

(Supplementary Fig. 7, Supplementary Table 1), which we used to build a complete model of the complex (Fig. 3a).

The structure of VvExoY is very similar to PaExoY. However, there are three striking differences. One concerns the N-terminal helix which is rotated by 90° in the VvExoY structure. Since it is located at the opposite side of the VvExoY-G-actin interface, an influence of this region on the actin-binding properties is unlikely. The other two differences are found in the ABD. First, the C-terminal helix of the anchor is extended by 6 residues in the

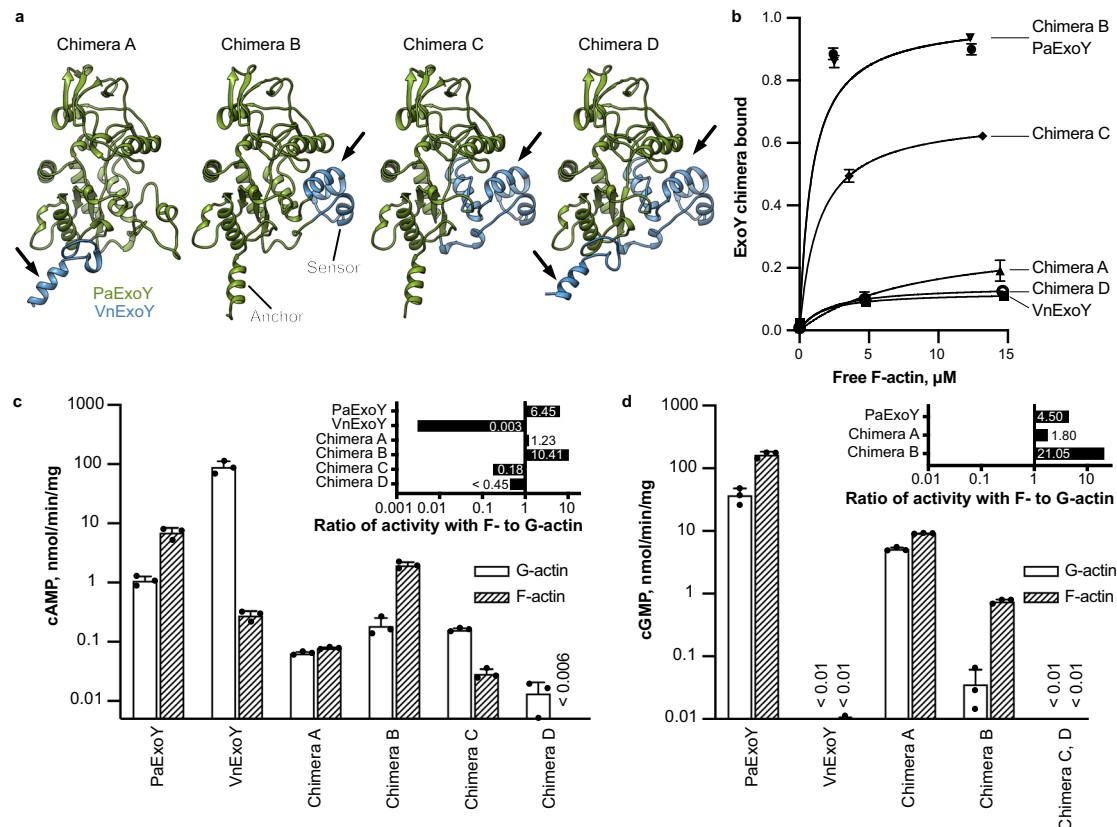

**Fig. 4 Chimera proteins of PaExoY and VnExoY. a** Models of chimera proteins, where regions of PaExoY (green) were swapped (arrows) with the corresponding parts of VnExoY (light blue) obtained by homology modeling. **b** Cosedimentations of F-actin with 2.5 µM of PaExoY, VnExoY or the chimera proteins quantified by densitometry. **c** Adenylyl and **d** Guanylyl cyclase activity of nucleotidyl cyclases in the presence of 2 µM latrunculin-stabilized G- or phalloidin-stabilized F-actin, measured during 60 min of incubation. All other chimera proteins created in this study are presented in Supplementary Fig. 8 and Supplementary Table 3. The data in **b–d**, are presented as mean values, the error bars correspond to standard deviations of three independent experiments. Individual data points are available in Source data.

case of VvExoY (Fig. 3b). Interestingly, this does not increase the interface with G-actin in comparison to the PaExoY-F-actin complex since the C-terminus of VvExoY protrudes away from actin.

The second difference in the ABD is more prominent, as the sensor of VvExoY is enlarged by 9 residues in comparison to PaExoY and rotated by ~90° allowing the ABD not only to establish contacts with subdomain I but also with subdomain II of G-actin. G-actin is in its expected globular conformation with a flexible D-loop. Since subdomain II is involved in intersubunit contacts in F-actin and parts of it are not accessible, the sensor of VvExoY would impair the binding to F-actin (Fig. 3b and c), explaining the preference of this toxin for G-actin.

**Alteration of the G-/F-actin specificity of ExoY toxins**. To corroborate that the ABDs are responsible for the selectivity of ExoY proteins for either F- or G-actin, we prepared PaExoY chimeras where we replaced parts of the ABD with the corresponding regions in VnExoY (Fig. 4, Supplementary Fig. 8, Supplementary Table 3). The construction of chimeras is in general a difficult task. Since these constructs often fail to express and function properly, we, therefore, prepared 15 different chimeras, of which 4 showed toxicity in yeast (Supplementary Fig. 8b). The change of the anchors resulted in a chimera (chimera A) that was equally activated by F- and G-actin (Fig. 4). The affinity to F-actin was considerably reduced in comparison to the wildtype.

Surprisingly, chimera B, where part of the sensor region that directly interacts with G-actin in VnExoY has been exchanged, still bound to F-actin and was preferably activated by the filament. This suggests that the altered sensor did not result in a steric clash with F-actin and corroborated that the anchor is very important for the selectivity of the toxin. When we exchanged the complete sensor plus a part of the catalytic center, the enzyme (chimera C) had a drastically reduced affinity to F-actin and strong preference for G-actin (Fig. 4c). Interestingly, similar to VnExoY, chimera C produced only cAMP but not cGMP, indicating that the change in the NBP altered the substrate specificity of the enzyme (Fig. 4c and d). When we exchanged both the anchor and sensor to the corresponding region of VnExoY, the resulting enzyme (chimera D) did not bind anymore to F-actin. Altogether, our structural and functional data on PaExoY and VvExoY complexes with actin and their chimeras show that the ABDs determine the selectivity of ExoY toxins for F- or G-actin. Thus, it should be possible to predict whether a certain ExoY is activated by F-actin or G-actin. To this end, we compared the sequences of all described ExoY-like proteins[15,22]. After removal of repetitive and incomplete sequences, we ended up with 25 unique ExoY-like proteins (Supplementary Fig. 9). We analyzed the differences and similarities in the ABDs of these proteins and divided them into four groups. Based on the similarity to PaExoY or VvExoY, the first two groups represent likely F-actin and G-actin-activated ExoY-like proteins with the corresponding sensor and anchor regions. While 9 ExoYs recognize G-actin, only the ExoY from *Aeromonas salmonicida*

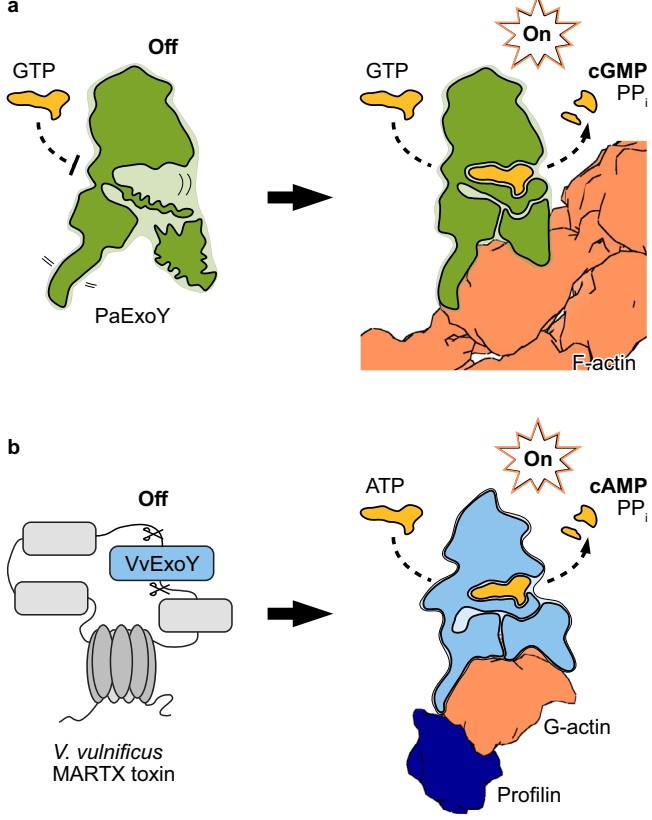

**Fig. 5 Activation of the actin-dependent nucleotidyl cyclase toxins. a** Formation of the PaExoY-F-actin complex results in an allosteric stabilization of the nucleotide binding pocket of PaExoY and thereby to an activation of the enzyme. **b** VvExoY is a module of MARTX toxins from *V. vulnificus*. After proteolytic cleavage, this module interacts with the G-actin-profilin complex and becomes a potent nucleotidyl cyclase. PPi inorganic pyrophosphate.

and PaExoY seem to be activated by F-actin. Interestingly, the two other groups, encompassing 14 unique sequences, possess completely different activator-binding regions, suggesting an existence of a new class of bacterial cyclase toxins that are neither activated by calmodulin nor actin.

## Discussion

Our results allow us to describe the activation mechanism of actin-activated nucleotidyl cyclase toxins in molecular detail (Fig. 5, Supplementary Movie 2). In the absence of an activator, the toxins are inactive due to the flexibility of large parts of the enzymes. After translocation of the toxins into their eukaryotic target cells either by T3SS in the case of PaExoY or as part of MARTX toxin in the case of ExoYs from *Vibrio* species, they bind to F-actin or G-actin. During this process, the highly flexible ABD becomes ordered on the surface of actin, resulting in the stabilization of the whole toxin including the NBP. Thus, the enzyme is activated by an allosteric activation-by-stabilization mechanism. In the case of *P. aeruginosa*, the activity of the toxin results in sharp increase of levels of cGMP and cAMP that hijacks cell signaling in the immune cells allowing the pathogen to circumvent the host immune response[9,10]. In the case of *V. vulnificus*, VvExoY produces cAMP which results in the disorganization of cell signaling and the promoting of inflammation in the infected area[13]. Differences in the sequence and conformation of the ABD are responsible for the selective binding to either F- or G-actin.

Notably, the specificity for the monomeric or filamentous actin can be biotechnologically modulated by exchanging the actin-binding regions from one toxin to the other. Based on our sequence analysis we identified a group of nucleotidyl cyclases that are likely neither activated by calmodulin nor actin. Future experiments will reveal the activator of these enzymes.

Other bacterial nucleotidyl cyclases, such as *B. anthracis* edema factor and CyaA from *B. pertussis* that bind to calmodulin rather than actin (Supplementary Fig. 3h), undergo a similar disordered-to-ordered transition during activation[16–18], suggesting that this mode of activation is conserved in these enzymes, albeit the different activator. As many human pathogens contain nucleotidyl cyclases as effectors, we expect that our findings allow to better understand the molecular mechanism of pathogenesis of infectious diseases and pave the road for the development of novel antidotes against poisons of microbial origin.

## Methods

**Plasmids, bacteria and yeast strains, growth conditions**. The list of the oligonucleotides, constructions and strains used in this study can be found in Supplementary Table 2. *E. coli* strains were cultivated in LB medium supplemented with kanamycin or ampicillin. *S. cerevisiae* were grown on rich YPD medium or synthetic defined medium (Yeast nitrogen base, Difco) containing galactose or glucose and supplemented if required with uracil, histidine, leucine, tryptophan, or adenine. *S. cerevisiae* strains were transformed using the lithium-acetate method[27]. Yeast viability upon toxin expression under the galactose promoter was analyzed by a drop test[28]. To this end, 5-fold serial dilutions of yeast suspensions normalized by $OD_{600}$ were spotted onto agar plates. Analysis of protein expression in yeast was performed by total protein extraction[29] (by incubating the cells in 0.1 M NaOH for 5 min and resuspending in Laemmli buffer), followed by SDS-PAGE, western blotting, and incubation with anti-myc (dilution 1:10000, 9B11 #2276, CST) or anti-RPS9 serum (dilution 1:10000, polyclonal rabbit antibodies were a generous gift of Prof. S. Rospert).

**Protein expression and purification**. *V. vulnificus* ExoY is expressed as a module of MARTX toxins, which are post-translationally cleaved inside the host cell. As in the article, we discuss exclusively the ExoY effector, we simplified the amino acid numbering by designating amino acid 3229 of the MARTX from *Vibrio vulnificus* strain BAA87 (GenBank: KJ131555.1) as the residue 1.

*Vibrio nigripulchritudo* ExoY, or fusion proteins of *P. aeruginosa* ExoY, *V. nigripulchritudo* ExoY, *V. vulnificus* ExoY or their chimeras, and maltose-binding protein (MBP) were purified from *E. coli* BL21-CodonPlus(DE3)-RIPL cells possessing the plasmid listed in Supplementary Table 2 according to the previously described protocol[12,30]. In short, a single colony was inoculated in 200 ml of LB media and grown at 37 °C until $OD_{600}$ of the bacterial suspension reached 1.0. Then, the protein expression was induced by addition of IPTG to 1 mM. After 2 h of expression at 37 °C, the cells were harvested by centrifugation and resuspended in buffer A (20 mM Tris-HCl pH 8, 500 mM NaCl). After ultrasonic lysis, the soluble fraction was loaded on Protino Ni-IDA resin, washed with buffer A, and eluted with buffer A, supplemented with 250 mM imidazole. After elution, recombinant ExoY proteins were dialyzed and stored in the buffer containing 20 mM Tris pH 8 and 150 mM NaCl.

Rabbit skeletal muscle alpha-actin was purified as described previously[31]. In short, rabbit muscle acetone powder (a generous gift of W. Linke and A. Unger, Ruhr-Universität Bochum, Germany) was resuspended in G-buffer (5 mM Tris-HCl, pH 7.5, 1 mM DTT, 0.2 mM CalCl2, 0.5 mM ATP). Then, the solution was centrifuged for 30 min at 100.000 × g to remove solid impurities and debris. The supernatant, containing G-actin, was mixed with MgCl2 and KCl to the final concentrations of 2 mM and 100 mM, respectively, to induce actin polymerization. After 1 h of incubation at room temperature, followed by addition of extra KCl to the final concentration of 800 mM to release actin-binding proteins, the actin filaments were centrifuged for 2 h at 100.000 × g. The protein pellet was then dialyzed against G-buffer for 2 days to depolymerize actin, and actin was polymerized and depolymerized once again. The final G-actin was stored in small aliquots at −80 °C. Human profilin-1 that was purified according to the described protocol[32] was a generous gift of Dr. J. Funk and Dr. P. Bieling. Thymosin β4 was purchased from Sigma-Aldrich (reference SRP3324).

Mammalian cytoplasmic beta-actin was expressed in C-terminal fusion with thymosin β4 and 10X-His-tag in insect cells BTI-Tnao38 following a similar strategy as described[33]. As Cys-272 was shown to be prone to oxidation in aqueous solutions[34] we introduced a Cys272Ala mutation, which is normally present in yeast actin. After baculovirus-mediated expression, cells were resuspended in the lysis buffer containing 10 mM Tris pH 8, 50 mM KCl, 5 mM CaCl2, 1 mM ATP, 0.5 mM TCEP and cOmplete protease inhibitor (Sigma), and lysed using a fluidizer. The supernatant after centrifugation was loaded on HisTrap FF crude (Thermo) equilibrated with lysis buffer. After washing step with the same buffer,

actin was eluted with a gradient of imidazole in the lysis buffer. After overnight dialysis in G-buffer (5 mM Tris pH 8, 2 mM CaCl$_2$, 0.5 mM ATP and 0.5 mM TCEP), actin was incubated with chymotrypsin for 20 min at 25 °C to remove thymosin β4 and the following His-tag. After addition of PMSF to the final concentration of 0.2 mM to stop the cleavage reaction, the mixture was applied onto the HisTrap FF column. The actin-containing flow-through was collected and polymerized overnight by addition of KCl and MgCl$_2$ to concentrations of 100 mM and 2 mM, respectively. The next day actin was spun down at 210,000 × g for 1 h. The pellet was resuspended in G-buffer and dialyzed for at least 3 days against G-buffer. Finally, the protein was spun down at 210,000 × g for 1 h, concentrated on 10 kDa cutoff Amicon columns, frozen in liquid nitrogen and stored at −80 °C.

**ExoY activity assays.** ExoY-dependent in vitro synthesis of cAMP or cGMP was measured according to the previously described method[15] with modifications to avoid use of radioactive materials. To start the reaction, 5 μl of 20 mM ATP or GTP solution was added to 45 μl mixture of ExoY, 2 mM MgCl$_2$ and G- or F-actin with phalloidin, latrunculin or actin-binding proteins, if indicated. After incubation at 30 °C, the reaction was stopped by the addition of 50 mm$^3$ of Al$_2$O$_3$ powder, which binds unreacted nucleoside triphosphates but does not interact with cyclic nucleotides. Following addition of 50 μl of TBS buffer, vortexing and centrifugation for 2 min at 15.000 × g, the light absorption of the supernatant was measured at the wavelength 252 or 259 nm for cGMP and cAMP, respectively. The amount of synthesized cNMP was then calculated using a calibration curve with defined concentration of cAMP or cGMP. In all reported experiments not more than 25% of the substrate was converted. The detection limit for each experiment was estimated as 0.1 optical density at the measured wavelength.

**Cosedimentation assays.** Affinity of the ExoY variants to F-actin was estimated using the high-speed cosedimentation assays, which were described previously[30]. To this end, we prepared 20 μl mixtures of F-actin and PaExoY at specified concentrations, incubated for 5 min at room temperature and spun down using the TLA-120 rotor at 150,000 × g for 20 min. Then, 5 μl of the supernatant and equal amount of the resuspended pellet was mixed with the Laemmli sample buffer and loaded on a stain-free TGX SDS-PAGE gel (Bio-Rad). Intensities of the ExoY bands were measured by densitometry in ImageLab, version 5.2.1. Dissociation constants were calculated in Prism, versions 8 and 9.

**Cryo-EM analysis of the PaExoY-F-α-actin complex.** Rabbit muscle F-actin was prepared as described previously[30]. In brief, the freshly thawed protein was spun down using the TLA-55 rotor for 30 min at 150,000 × g at 4 °C, and the G-actin-containing supernatant was collected. Then, the protein was polymerized by incubation in the buffer containing 120 mM KCl, 20 mM Tris pH 8, 2 mM MgCl$_2$, 1 mM DTT, and 1 mM ATP (F-buffer) in the presence of a twofold molar excess of phalloidin for 30 min at room temperature and further overnight at 4 °C. The next day, the actin filaments were pelleted using the same TLA-55 rotor for 30 min at 150,000 × g at 4 °C and resuspended in F-buffer. Fifteen minutes before plunging, F-actin was diluted to 2 μM, mixed with 4 μM of PaExoY, 2 mM 3′-deoxyguanosine-5′-triphosphate (3′-dGTP, Jena Bioscience) and 4 mM MgCl$_2$. Shortly before plunging, Tween-20 was added to the sample to a final concentration of 0.02% (w/v) to improve the ice quality. Plunging was performed using the Vitrobot Mark IV system (Thermo Fisher Scientific) at 13 °C and 100% humidity. 3 μl of sample was applied onto a freshly glow-discharged copper R2/1 300 mesh grid (Quantifoil), blotted for 8 s on both sides with blotting force −20 and plunge-frozen in liquid ethane.

The dataset was collected using a Krios Titan transmission electron microscope (Thermo Fisher Scientific) equipped with an XFEG at 300 kV using the automated data-collection software EPU, version 2.7 (Thermo Fisher Scientific). Five images per hole with a defocus range of −0.4 to −3.5 μm were collected with the Falcon III detector (Thermo Fisher Scientific) operated in linear mode. Image stacks with 40 frames were collected with a total exposure time of 1.5 s and total dose of 93 e$^−$/Å-. 12437 images were acquired and 8663 of them were used for processing. Motion correction and CTF estimation were performed in CTFFIND[35], version 4.1.1, and MotionCorr2[36], version 1.1.0, during image acquisition with TranSPHIRE[37], version 1.4.28. Filament picking was performed using crYOLO[38] version 1.5. On the next step, 2.25 million helical segments were classified in 2D using ISAC[39] in Sphire version 1.3 to remove erroneous picks. The remaining 1.85 million particles were used in the first 3D refinement in Meridien[40] in Sphire version 1.3 with 25 Å low-pass filtered F-actin map as the initial model and with a spherical mask with a diameter of 308 Å, followed by a local 3D refinement with a wide mask of the shape of the PaExoY-F-actin complex. On the next step, per-particle CTF-refinement was performed using 3D refinement parameters and the 3D reconstruction in Sphire, version 1.3, followed by removal of the segments with defocus value lower than −3 μm and recentering of the remaining 1.71 million particles according to the parameters calculated in the previous 3D refinement. Following the third round of 3D refinement in Sphire, Bayesian polishing and 3D classification were performed in Relion version 3[41]. The latter step was performed with a mask covering three actin and one PaExoY subunit in order to remove particles that do not possess bound PaExoY in the particular position. Indeed, 176067 segments without PaExoY density were removed and the remaining 1.54 million were introduced

into the final round of 3D refinement with a tight mask covering actin and PaExoY subunits. The final reconstruction map was postprocessed using 3D local filter based on the local resolution of the map in Sphire. Maps presented on Fig. 1a and Supplementary Movie 2 were postprocessed using DeepEMhancer version 1.0[42]. The overall processing, FSC curves and local resolution maps are available in Supplementary Fig. 1.

To build a model of the PaExoY-F-actin complex, we performed flexible fitting of the PaExoY partial crystal structure (5XNW[22]) into the EM density using iMODFIT Chimera plugin[43], version 1.2. Then, we build the remaining 106 amino acids using Rosetta software version 3[44,45] and added 3′-dGTP, which was modeled with eLBOW[46]. F-actin, ADP-$_\text{Pi}$ and Phalloidin were adapted from PDB 7AD9[30]. The model was further refined using ISOLDE[47], version 1.0B4, and Phenix[48], version 1.17. Schematic representation of the secondary structure elements of PaExoY for Fig. 1c was calculated in UCSF Chimera, version 1.14, with H-bond energy cutoff −0.45 kcal/mol.

The Buried surface area of the PaExoY-F-actin interface and distances between ligands and neighboring atoms (Supplementary Fig. 2i) were calculated using PDBsum[49].

**Cryo-EM analysis of the VnExoY-G-α-actin complex.** Freshly thawed rabbit muscle actin was spun down using TLA-55 rotor for 30 min at 150,000 × g at 4 °C, and the supernatant with G-actin was collected. Three μl of the sample containing 2.2 μM of VnExoY, 2 mM 3′-deoxyadenosine-5′-triphosphate (3′-dATP, Jena Bioscience), 4 mM MgCl$_2$ and 2 μM G-actin, were applied onto a freshly glow-discharged copper R2/1 300 mesh grid (Quantifoil), blotted for 3 s on both sides with a blotting force −3 and plunge-frozen in liquid ethane using the Vitrobot Mark IV system (Thermo Fisher Scientific) at 13 °C and 100% humidity.

The data collection was performed on a Krios Titan transmission electron microscope (Thermo Fisher Scientific) equipped with an XFEG at 300 kV and CS-corrector using the automated data-collection software EPU, version 2.7 (Thermo Fisher Scientific). Four images per hole with a defocus range of −1 to −2.5 μm were collected with a Gatan K2 camera operated in counting mode. Image stacks with 64 frames were collected with a total exposure time of 8 s and a total dose of 80 e$^−$/Å$^2$. 10,659 images were acquired and 9329 of them were used for processing. Motion correction and CTF estimation have been similarly performed in CTFFIND[35], version 4.1.13, and MotionCorr2[36], version 1.2.6, during image acquisition using TranSPHIRE[37], version 1.4.28. Particle picking using crYOLO[38], version 1.6, resulted in 1.86 million particles, which were used in 3D refinement in Meridien in Sphire, version 1.3. For the latter, a 25 Å low-pass filtered map of a single subunit of the PaExoY-F-actin complex was used as an initial model. After application of 2D shifts for recentering of the particles and removal of overrepresented views using Sphire, 2D classification was used to remove erroneous picks. The selected 1.1 million particles were subjected to three rounds of 3D refinement with particle restacking and particle polishing in Relion[41] version 3 in between. After the second polishing step and another round of 2D classification, 3D refinement was performed with a tighter mask and initial model from the previous 3D refinement output. After the third round of particle polishing, the final round of 3D refinement was performed using Sidesplitter filter to reduce local overfitting due to the preferred orientation of the particles in ice[50]. The final reconstruction map was evaluated with 3D FSC tool[51] and postprocessed using DeepEMhancer, version 1.0[42]. The processing overview with intermediate maps, angular distribution graphs, and FSC curves is available in Supplementary Fig. 6.

**Cryo-EM analysis of the VvExoY-G-β-actin-profilin complex.** Freshly thawed mammalian beta actin was spun down using the TLA-120 rotor for 20 min at 120,000 × g at 4 °C, and the supernatant with G-actin was collected. A mixture containing G-actin at 18 μM, human profilin-1 at 24.5 μM, MBP-VvExoY at 18 μM, 3′-deoxyadenosine-5′-triphosphate (3′-dATP, Jena Bioscience) at 2 mM and MgCl$_2$ at 4 mM was incubated for 15 min at room temperature. Shortly before plunging, the sample was diluted 5 times with 10 mM Tris pH 8 supplemented with 0.002% Tween-20. Then, the mixture was applied onto a freshly glow-discharged copper R2/1 300 mesh grid (Quantifoil), blotted for 3 s on both sides with a blotting force −5 and plunge-frozen in liquid ethane using the Vitrobot Mark IV system (Thermo Fisher Scientific) at 13 °C and 100% humidity.

The data collection was performed on a Krios Titan transmission electron microscope (Thermo Fisher Scientific) equipped with an XFEG at 300 kV and CS-corrector using the automated data-collection software EPU, version 2.8 (Thermo Fisher Scientific). Three images per hole with a defocus range of −1.2 to −2.5 μm were collected with a Gatan K3 camera operated in superresolution mode. Image stacks with 60 frames were collected with a total exposure time of 2 s and a total dose of 60 e$^−$/Å$^2$. Following acquirement of 3773 movies without stage tilt, 3106 micrographs were imaged in an equal number using 10-, 20- or 30-degrees stage tilt. After visual inspection, 378 non-tilted and 183 tilted micrographs were removed. For the remaining images, motion correction and CTF estimation have been performed in CTFFIND[35], version 4.1.13, and MotionCorr2[36], version 1.3, during image acquisition using TranSPHIRE[37], version 1.5.13. Particle picking using crYOLO[38], version 1.8, resulted in 706 thousand non-tilted and 543 thousand tilted particles that were used in independent 3D refinements in Meridien in Sphire version 1.4. For the latter, a 25 Å low-pass filtered map of the simulated VnExoY-G-actin complex with docked profilin (PDB 2PAV) was used as the initial

model. After the application of 2D shifts for recentering of the particles, 2D classification was used to remove erroneous picks. The selected 222 and 130 thousand particles, non-tilted and tilted, respectively, were subjected to the second round of 3D refinement. Using the 2D shift parameters obtained in the 3D refinement step, the particles were recentered and merged into one stack. After two more rounds of 3D refinement and particle polishing in Relion, version 3.1, in between[41], the resulting 3.9 Å map was obtained and used for modeling of the VvExoY-G-actin complex. To compare the quality of the map with the density of the VnExoY-G-actin complex, the map was additionally evaluated with 3D FSC tool[51].

To further boost the profilin density, we performed an additional alignment-free 3D classification in Relion, version 3.1, followed by 3D refinement in Meridien in Sphire version 1.4. The resulting 4.2 Å map was used to model the actin-profilin interface. The processing overview with intermediate maps, angular distribution graphs, and FSC curves is available in Supplementary Fig. 7.

To build a model of the VvExoY-G-actin-profilin complex, we first modeled VvExoY using tFold server (https://drug.ai.tencent.com/) and fit it into the EM density using iMODFIT Chimera plugin, version 1.2[43]. Amino acids 1-38, 236-273, 399-432 that did not fit the experimental density were removed and remodeled using Rosetta software, version 3[44,45]. The VvExoY model was then merged with profilin-G-actin complex from PDB 6NBW[52] and further refined using ISOLDE, version 1.0B4[47] and Phenix, version 1.17[48].

**Homology modeling**. Homology models of the chimeric ExoY proteins were built using MODELLER[53]. For this, we used PaExoY and VvExoY as templates, but deleting the areas of the protein that should not be taken into account (e.g., PaExoY's C-terminus in Chimera A).

**Molecular dynamics simulations**. We used the cryo-EM model of the PaExoY-F-actin complex to set up classical molecular dynamics simulations of PaExoY in the presence or absence of F-actin. For free PaExoY, we took the coordinates of a single chain from the decorated filaments and solvated it with TIP3P waters 15 Å away from the furthest protein atom, using a truncated octahedron box. For the PaExoY-F-actin complex, we built a system composed of a single PaExoY molecule bound at the center of a filament made of 9 actin subunits. We then placed the complex into a TIP3P water box of dimensions 150×150×315 Å. KCl was then used to neutralize the systems and bring the ionic strength to 150 mM. All N-termini were acetylated. In simulations 1 and 2 of the F-actin complex, C-terminus of PaExoY was methylamidated, while simulations 3 and 4 had a standard C-terminus. All simulations were prepared in CHARMM-GUI[54] using the CHARMM 36 m forcefield[55]. Simulations were run in NAMD 2.14[56], using a time step of 2 fs. All bonds involving hydrogens were constrained to their equilibrium length. Short-range non-bonded interactions were truncated at 12 Å, with switching starting at 10 Å. Long-range electrostatics were treated using particle mesh Ewald[57]. We performed 2 independent simulations of free PaExoY, and 4 of the F-actin-PaExoY complex, all started from random velocities. The systems were equilibrated using the stepwise protocol summarized in Supplementary Table 4, and then run for 400 ns.

Simulation results were analyzed using VMD[58], version 1.9.4 alpha and Carma[59], version 2.01. The network analysis was performed using the NetworkView plugin of VMD[26].

**Reporting summary**. Further information on research design is available in the Nature Research Reporting Summary linked to this article.

## Data availability

The coordinates for the cryo-EM structures of PaExoY-F-actin, VvExoY-G-actin-profilin, and VnExoY-G-actin have been deposited in the Electron Microscopy Data Bank under accession numbers EMD-13158, EMD-13159, EMD-13160. The corresponding molecular models for the PaExoY-F-actin, VvExoY-G-actin-profilin complexes have been deposited at the wwPDB with accession codes PDB 7P1G, 7P1H. The raw data generated during the current study are available from the corresponding author on reasonable request. Individual data points from the graphs at Fig. 4b–d, Supplementary Figs. 2e, g, 3b, d, g, and 5 are available in Source data. Uncropped gels and western blots from Supplementary Figs. 2d, f, 3e, f, and 8c can be found in Supplementary Fig. 10. Source data are provided with this paper.

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

## Acknowledgements

We thank O. Hofnagel and D. Prumbaum for assistance with data collection and maintaining the EM facility, S. Bergbrede for the excellent technical assistance, W. Linke and A. Unger (Ruhr-Universität Bochum, Germany) for providing us with muscle acetone powder, S. Rospert (University of Freiburg, Germany) for providing us anti-RPS9 serum, J. Funk and P. Bieling for providing us with profilin-1, and S. Pospich, Y. Belyi, and D. Ladant for fruitful discussions. This work has been funded by the Max Planck Society (S.R.). U.M. was supported by ANR (Agence nationale de la recherche) grant ANR-18-CE44-0004. A.B. was supported by a fellowship of the Humboldt foundation and an EMBO long-term fellowship.

## Author contributions

A.B. and S.R. designed the project. A.B. prepared cryo-EM specimens, collected and analyzed EM data, performed affinity and activity assays. A.B. built the atomic models with support from F.M. F.M. performed and analyzed MD simulations. A.B. and F.M. prepared figures. S.R. supervised the project. A.B., F.M. and S.R. wrote the manuscript with input from U.M.

## Funding

## Competing interests

The authors declare no competing interests.
