## [Peer Review File · Nature Communications]

Mechanism of actin-dependent activation of nucleotidyl cyclase toxins from bacterial human pathogensREVIEWER COMMENTS

Reviewer #1 (Remarks to the Author):

Belyi and co-workers report here a quite complete and very carefully executed study that represents a lovely piece of structural biology work. It provides an unprecedented mechanistic insight into the molecular details of structuring of the catalytic sites and mode of activation of two prototypic PaExoY and VvExoY nucleotidyl cyclase toxin domains from two important bacterial pathogens *Pseudomonas aeruginosa* and *Vibrio vulnificus*, respectively. For the first time the atomic details of an actin-activated nucleotidyl cyclase are reported. The high resolution (3.2 and 3.6 angstrom) Cryo-EM structures, of PaExoY in complex with F actin and VvExoY in complex with G actin/profilin, combined with the mutagenesis work, molecular dynamics simulations and enzymatic activity assays, and complemented with the very elegant approach of constructing molecular and activation analysis of chimeras of the ExoY proteins, this all in sum enabled the authors to decipher the detailed structural mechanism of selectivity of binding and activation of the two NMP cyclase enzymes by the filamentous (F) and soluble (G) forms of actin, respectively. Sequence homology alignments and modeling then allowed predicting of a third class of yet unknown eukaryotic cytosolic activators of a novel class of ExoY homologues.

The reported results are highly novel, original, of high quality and are important for the entire bacterial pathogenesis and toxin research field. The presented data constitute a new paradigm of structure-function relationship and ABS and NBS domain organization and formation of bacterial toxin nucleotidyl cyclase enzymes, with conclusions allowing generalization.

The work is very carefully executed and very thoroughly described when it comes to data and important technical details of protein preparation and structure solving. Some elegant tricks were used, such as involving profilin in G actin complex formation for structure solving of VvExoY, taking advantage of G actin capturing by profiling *in vivo*.

The manuscript is carefully and very clearly written and despite of the succinct style, it is generally well understandable and is packed with wealth of important and original information, thus making it a true delight to read. The presented data do well support the claims and conclusions and the technical performance is of very high standard, as is the figure preparation and description.

The supplementary materials are carefully prepared and very useful for complementing the main text. There is hardly anything to criticize on this paper, which is rather rare. Congratulations!

I only have minor spelling/wording suggestions, as outlined below:

- a) p. 1 – Abstract, 2nd sentence contains an important logical shortcut that is far from being obvious to understood by readers that are not experts in activation of bacterial toxin nucleotidyl cyclase enzymes. These will not know that the toxic bacterial enzymes are inactive inside bacterial cytosol and get activated only once delivered into host target cell cytosol by binding of some eukaryotic cytosolic protein(s). Consider developing/rewording the sentence for better understanding by the general readership of the journal.
- b) p. 2, l. 10 - ...death of the intoxicated cultured cells...
- c) p. 2, l.10/11 – which are more likely to occur during... (verb missing)
- d) p. 2, 2nd paragraph - ... module of a MARTX...
- e) p. 2, 3rd paragraph - ...such as the edema... and the adenylate...
- f) p. 11 - ... P. Bieling and...
- g) p. 12 – ExoY activity assays – ...Al₂O₃ powder that binds...
- h) p. 13, 14, 15 – protein/actin was spun... (not span...)

Reviewer #2 (Remarks to the Author):

In this paper by Belyy et al., the authors lay out a structural mechanism of actin-mediated enzyme activation of nucleotidyl cyclase toxins in bacterial human pathogens. A remarkable observation made here is that binding of these bacterial enzymes to actin (in either filamentous or monomer forms) allosterically modifies the active site to activate the cyclase function. For the enzyme studied in the most detail (PaExoY), mutational analysis is used to identify key interactions involved in the allosteric activation pathway, and molecular dynamics simulations (together with an established tool, dynamic network analysis) are used to suggest specific residues involved in the allosteric communication. A final part of this study involves an ambitious attempt to switch the functionality of the F-actin binding PaExoY enzyme by transplanting structural elements from the G-actin binding cousin, VvExoY. While the activity of the resulting chimera constructs was reduced quite substantially, one or possibly two of the constructs did achieve a reversal in selectivity. These results provide a structural basis for understanding how these important bacterial toxins specifically target the host and also how different forms of actin can be targeted. Implications for similar behavior of other bacterial toxins are discussed. Overall, the manuscript is well written and contains high impact results which may advance the understanding of interactions between actin and various actin-binding proteins in general. Below are a few suggestions and a couple minor corrections.

1) The functional significance of the PaExoY sensor's salt bridge with F-actin [D247 — R95] is well established by the mutational analysis presented, and I assume the MD simulations backed this up as D247 was itself used for the network analysis, under the assumption that this interaction is important for allostery. However, it would be nice to see direct support from the MD for the stability of this salt bridge- i.e. is it maintained for the duration of the simulation (or even a PMF plot where the reaction coordinate is the salt bridge formation).

2) Apart from the salt bridge, are there any other interactions seen between sensor and F-actin in the simulations preserved through the trajectory?

3) 'Altogether, our structural and functional data on PaExoY and VvExoY complexes with actin and their chimeras show that the ABDs determine the selectivity of ExoY toxins for F- or G-actin.' — This somewhat overstates the case because the chimeras all substantially degraded the activity of the enzyme, thus leaving it somewhat unclear whether other parts of the enzyme participate in the recognition mechanism. It would be good if the authors can comment on this. It seems likely that reduction of function in the chimeras was due to partial disruption of allosteric pathways from actin to the active site- does the dynamic network analysis provide any insight into how this could happen?

4) The authors don't expand on VnExoY anchor domain, can they comment on whether the anchor residues in VnExoY are maintained and conserved as seen in PaExoY?

5) The idea of 'sensor' and 'anchor' suggested in the first part of the paper (and nicely supported by mutagenesis and affinity data) becomes muddled in the second part of the paper when the chimeras make it clear that both 'anchor' and 'sensor' are involved in discriminating G from F actin. In other words, the 'sensor' is not a 'G/F actin sensor' as a reader might initially think. It could be helpful for the flow of the text to add a comment in the discussion to this effect; in other words, address why it was originally suggested that sensor and anchor domains possess distinct functions, but that both seem to contribute to conformational recognition of the actin substrate.

Corrections:

1) Under the section "Alteration of G-/F-actin specificity of ExoY toxins" there is a typo: "To corroborate that..., we prepared PaExoY chimeras where we replaced parts of the ABD".

2) In Fig S3A, the legend is mislabeled. It needs to be 'root mean square deviation' as opposed to 'root mean square fluctuation'.

Reviewer #3 (Remarks to the Author):

The manuscript by Raunser and colleagues "Mechanism of actin-dependent activation of nucleotidyl cyclase toxins from bacterial human pathogens" presents first cryoEM structures of important members of this family, notably from *Pseudomonas* and *Vibrio* pathogens, bound to actin with several supporting experiments to validate a mechanism of activation therein. Building on prior crystallographic structures of the apo form of the toxins, the well done and well presented work here fills in missing regions now shown to be critical for actin binding and activation. I believe this work will be of significant interest to the broad academic and pharmaceutical microbial pathogenesis sectors interested in host/pathogen interactions and toxin action.

Suggested minor revisions for publication:

Last statement of the abstract should be toned down/removed. May not apply to the whole family as the authors themselves state on Page 8, last line?

Please address use of Phalloidin for the reader (intro or results section – what is this compound and known effects, why needed here, could its presence skew interaction with the toxin?).

Ditto for profilin.

Pg. 3 – top – please provide any missing residue ranges in the final model in the text (re statement "allowed us to build an almost full atomic model of the complex"). This is not addressed in the figures/Table cited there.

Pg. 3 – last paragraph - please provide an RMSD on common atoms between apo and actin bound forms here

Brief description supporting relevance of use of the yeast toxicity assay

Pg. 4, 3rd paragraph - Please provide buried surface area and binding constant of WT toxin and actin (after "anchor largely determines the affinity").

Pg. 5 top – Please provide here or in a supp figure the map region overlaid on and defining the bound nucleotide and described magnesiums.

Please provide the coordination geometry and ligand distances for the Mg²⁺'s

Pg. 5 bottom – Could you verify the MD predicted transition with differential CD or other spectroscopic methods?

There are grammatical errors throughout and the manuscript would benefit from a careful editing.

We thank the reviewers for their positive and constructive feedback, which aided us to further improve the manuscript. Major modifications of the manuscript are highlighted in yellow. Below we include our detailed response to each point raised.

REVIEWER COMMENTS

Reviewer #1 (Remarks to the Author):

Belyi and co-workers report here a quite complete and very carefully executed study that represents a lovely piece of structural biology work. It provides an unprecedented mechanistic insight into the molecular details of structuring of the catalytic sites and mode of activation of two prototypic PaExoY and VvExoY nucleotidyl cyclase toxin domains from two important bacterial pathogens *Pseudomonas aeruginosa* and *Vibrio vulnificus*, respectively. For the first time the atomic details of an actin-activated nucleotidyl cyclase are reported. The high resolution (3.2 and 3.6 angstrom) Cryo-EM structures, of PaExoY in complex with F actin and VvExoY in complex with G actin/profilin, combined with the mutagenesis work, molecular dynamics simulations and enzymatic activity assays, and complemented with the very elegant approach of constructing molecular and activation analysis of chimeras of the ExoY proteins, this all in sum enabled the authors to decipher the detailed structural mechanism of selectivity of binding and activation of the two NMP cyclase enzymes by the filamentous (F) and soluble (G) forms of actin, respectively. Sequence homology alignments and modeling then allowed predicting of a third class of yet unknown eukaryotic cytosolic activators of a novel class of ExoY homologues.

The reported results are highly novel, original, of high quality and are important for the entire bacterial pathogenesis and toxin research field. The presented data constitute a new paradigm of structure-function relationship and ABS and NBS domain organization and formation of bacterial toxin nucleotidyl cyclase enzymes, with conclusions allowing generalization.

The work is very carefully executed and very thoroughly described when it comes to data and important technical details of protein preparation and structure solving. Some elegant tricks were used, such as involving profilin in G actin complex formation for structure solving of VvExoY, taking advantage of G actin capturing by profiling in vivo.

The manuscript is carefully and very clearly written and despite of the succinct style, it is generally well understandable and is packed with wealth of important and original information, thus making it a true delight to read. The presented data do well support the claims and conclusions and the technical performance is of very high standard, as is the figure preparation and description.

The supplementary materials are carefully prepared and very useful for complementing the main text. There is hardly anything to criticize on this paper, which is rather rare. Congratulations!

I only have minor spelling/wording suggestions, as outlined below:

- a) p. 1 – Abstract, 2nd sentence contains an important logical shortcut that is far from being obvious to understood by readers that are not experts in activation of bacterial toxin nucleotidyl cyclase enzymes. These will not know that the toxic bacterial enzymes are inactive inside bacterial cytosol and get activated only once delivered into host

target cell cytosol by binding of some eukaryotic cytosolic protein(s). Consider developing/rewording the sentence for better understanding by the general readership of the journal.

We have changed this passage accordingly.

- b) p. 2, l. 10 - ...death of the intoxicated cultured cells...
- c) p. 2, l.10/11 – which are more likely to occur during... (verb missing)
- d) p. 2, 2nd paragraph - ... module of a MARTX...
- e) p. 2, 3rd paragraph - ...such as the edema... and the adenylate...
- f) p. 11 - ... P. Bieling and...
- g) p. 12 – ExoY activity assays – ...Al₂O₃ powder that binds...
- h) p. 13, 14, 15 – protein/actin was spun... (not span...)

In general, we thank the reviewer for the very positive feedback on our manuscript. We have changed all the typos and grammar issues in the revised manuscript.

Reviewer #2 (Remarks to the Author):

In this paper by Belyy et al., the authors lay out a structural mechanism of actin-mediated enzyme activation of nucleotidyl cyclase toxins in bacterial human pathogens. A remarkable observation made here is that binding of these bacterial enzymes to actin (in either filamentous or monomer forms) allosterically modifies the active site to activate the cyclase function. For the enzyme studied in the most detail (PaExoY), mutational analysis is used to identify key interactions involved in the allosteric activation pathway, and molecular dynamics simulations (together with an established tool, dynamic network analysis) are used to suggest specific residues involved in the allosteric communication. A final part of this study involves an ambitious attempt to switch the functionality of the F-actin binding PaExoY enzyme by transplanting structural elements from the G-actin binding cousin, VvExoY. While the activity of the resulting chimera constructs was reduced quite substantially, one or possibly two of the constructs did achieve a reversal in selectivity. These results provide a structural basis for understanding how these important bacterial toxins specifically target the host and also how different forms of actin can be targeted. Implications for similar behavior of other bacterial toxins are discussed. Overall, the manuscript is well written and contains high impact results which may advance the understanding of interactions between actin and various actin-binding proteins in general. Below are a few suggestions and a couple minor corrections.

- 1) The functional significance of the PaExoY sensor's salt bridge with F-actin [D247 — R95] is well established by the mutational analysis presented, and I assume the MD simulations backed this up as D247 was itself used for the network analysis, under the assumption that this interaction is important for allostery. However, it would be nice to see direct support from the MD for the stability of this salt bridge- i.e. is it maintained for the duration of the simulation (or even a PMF plot where the reaction coordinate is the salt bridge formation).
- 2) Apart from the salt bridge, are there any other interactions seen between sensor and F-actin in the simulations preserved through the trajectory?

During the revision of the manuscript, we noticed that in the simulations of PaExoY-F-actin, PaExoY had its C-terminus methylamidated. Although experimentally we know that deletion

of the last three amino acids does not change toxicity of PaExoY in a yeast model (Belyy et al., JBC 2018), we performed two new simulations with a standard C-terminus. We could not identify any significant difference (see below RMSF and RMSD in Fig S3A) and therefore decided to combine all four in the manuscript.

The D247-R95 salt-bridge is indeed very well conserved in all simulations we performed. As correctly pointed out by the reviewer, interactions are only considered in the network if they are present 75% of the time. We believe that the stability of this particular interaction is in part a result of the bridge being somewhat buried within the interface. As requested by the reviewer, we added a plot of the $R95_{CZ}$ - $D247_{CG}$ to the supplementary figure S4 to illustrate this. There aren't many more specific interactions with the sensor. Nevertheless, L243 is buried within a hydrophobic pocket created by the tail of R95, A97 and Y91. We highlighted those as well (Fig S4). We also included the trajectories for two interactions of D25, as we know this actin residue is key for PaExoY's binding. We think that a PMF calculation for the D247-R95 salt-bridge formation would be out of the scope of this paper, as a detailed thermodynamic characterization of the formation of this interaction is unlikely to further our understanding of the structural mechanism of PaExoY's activation. In addition, our systems are composed of ~ 670.000 atoms and this calculation would become exceedingly long for our revision time.

3) 'Altogether, our structural and functional data on PaExoY and VvExoY complexes with actin and their chimeras show that the ABDs determine the selectivity of ExoY toxins for F- or G-actin.' — This somewhat overstates the case because the chimeras all substantially degraded the activity of the enzyme, thus leaving it somewhat unclear whether other parts of the enzyme participate in the recognition mechanism. It would be good if the authors can comment on this.

This reviewer is right in stating that the activity of the chimeras was reduced which could in part be due to a partly distorted activation pathway and we that cannot fully rule out that other regions of the enzyme contribute to the recognition mechanism. We therefore toned down our statement and write instead that the ABDs are key for the selectivity although we cannot rule out that the other regions of the enzyme contribute to the recognition mechanism.

It seems likely that reduction of function in the chimeras was due to partial disruption of allosteric pathways from actin to the active site- does the dynamic network analysis provide any insight into how this could happen?

Although it is in principle likely that internal communication pathways are interrupted in the chimeras, extrapolation from our current network analysis would be somehow misleading. The communication of the allosteric signal will depend on both, how the grafted regions interact with actin and how they are internally “wired”, none of which could be guessed in a straightforward manner from our simulations.

4) The authors don't expand on VnExoY anchor domain, can they comment on whether the anchor residues in VnExoY are maintained and conserved as seen in PaExoY?

We have forgotten to name the anchor in the comparison. This has been done in the revision.

5) The idea of ‘sensor’ and ‘anchor’ suggested in the first part of the paper (and nicely supported by mutagenesis and affinity data) becomes muddled in the second part of the paper when the chimeras make it clear that both ‘anchor’ and ‘sensor’ are involved in discriminating G from F actin. In other words, the ‘sensor’ is not a ‘G/F actin sensor’ as a reader might initially think. It could be helpful for the flow of the text to add a comment in the discussion to this effect; in other words, address why it was originally suggested that sensor and anchor domains possess distinct functions, but that both seem to contribute to conformational recognition of the actin substrate.

We thank the reviewer for this comment and are sorry for the confusion. Indeed, we termed actin-binding regions as ‘sensor’ and ‘anchor’ to discriminate their functions and provide the reader with the easy-to-understand naming. Instead of being a G/F actin sensor, the region we call sensor senses the presence of the activator, i.e. G- or F-actin alike. We changed the description in the text to avoid any further possible confusion.

Corrections:

1) Under the section “Alteration of G-/F-actin specificity of ExoY toxins” there is a typo: “To corroborate that..., we prepared PaExoY chimeras where we replaced parts of the ABD”.

We corrected the text as suggested by the reviewer.

2) In Fig S3A, the legend is mislabeled. It needs to be ‘root mean square deviation’ as opposed to ‘root mean square fluctuation’.

We thank the reviewer for noticing this. We changed RMSF to RMSD.

Reviewer #3 (Remarks to the Author):

The manuscript by Raunser and colleagues “Mechanism of actin-dependent activation of nucleotidyl cyclase toxins from bacterial human pathogens” presents first cryoEM structures of important members of this family, notably from *Pseudomonas* and *Vibrio* pathogens, bound to actin with several supporting experiments to validate a mechanism of activation therein. Building on prior crystallographic structures of the apo form of the toxins, the well done and well presented work here fills in missing regions now shown to be critical for actin binding and activation. I believe this work will be of significant interest to the broad academic and

pharmaceutical microbial pathogenesis sectors interested in host/pathogen interactions and toxin action.

Suggested minor revisions for publication:

Last statement of the abstract should be toned down/removed. May not apply to the whole family as the authors themselves state on Page 8, last line?

We toned down this statement as suggested.

Please address use of Phalloidin for the reader (intro or results section – what is this compound and known effects, why needed here, could its presence skew interaction with the toxin?).

Ditto for profilin.

We added changes to the text to explain the rationale of using phalloidin in our studies. In our previous publications we demonstrated that phalloidin does not change the affinity of the PaExoY-F-actin complex (Belyy *et al.*, Nat Commun 2016, Belyy *et al.*, JBC 2018).

Concerning profilin, the information is already present in the main result text and supplemented by an experiment presented in the Supplementary Figure 5.

Pg. 3 – top – please provide any missing residue ranges in the final model in the text (re statement “allowed us to build an almost full atomic model of the complex”). This is not addressed in the figures/Table cited there.

Thank you for pointing it out. We added this information in Supplementary Table 1.

Pg. 3 – last paragraph - please provide an RMSD on common atoms between apo and actin bound forms here

As requested, we provided RMSD on common atoms in the text.

Brief description supporting relevance of use of the yeast toxicity assay

As requested by the reviewer, we added several sentences supporting the relevance of the yeast assay in our study.

Pg. 4, 3rd paragraph - Please provide buried surface area and binding constant of WT toxin and actin (after “anchor largely determines the affinity”).

As requested by the reviewer, we provided the buried surface area of the interface between the anchor of PaExoY and actin and specified the binding constant of the toxin and F-actin.

Pg. 5 top – Please provide here or in a supp figure the map region overlaid on and defining the bound nucleotide and described magnesiums.

As suggested by the reviewer, we added Supplementary figure 2H,I to show the bound nucleotide and magnesiums fitted into the EM map.

Please provide the coordination geometry and ligand distances for the Mg²⁺'s

As requested by the reviewer, we added Supplementary figure 2I showing distances between the Mg²⁺ ligand and neighboring atoms. However, since the local resolution of this area is around 3.3-3.5 Å and we do not see any surrounding water molecules, the position of the coordinating ligands can, at most, be inferred from homologous structures. This is also the case for the identity of the ions themselves.

Pg. 5 bottom – Could you verify the MD predicted transition with differential CD or other spectroscopic methods?

We thank the reviewer for the interesting suggestion. Indeed, it would be nice to demonstrate F-actin induced stabilization of PaExoY additionally by a spectroscopic method.

As suggested by the reviewer, we started with circular dichroism. This method is commonly used to investigate the changes in secondary structure upon complex formation. However, in our case, the MD simulations of PaExoY (Supplementary Movie 1) predicted changes in the tertiary structure only. Nevertheless, we acquired the spectra of 2 μM PaExoY, 2 μM F-actin and 2 μM PaExoY-F-actin complex. To detect any differences, we calculated the sum of the spectra of PaExoY and F-actin alone and subtracted it from the spectrum of PaExoY-F-actin (see figure below). However, we did not see any significant difference between the spectra, suggesting that there is no large change of the secondary structure upon complex formation, corroborating our MD simulations. Therefore, we conclude that we cannot detect the conformational changes during PaExoY activation using this method.

Then, as suggested by the reviewer, we decided to try another spectroscopic method. According to our dynamic network analysis, upon interaction with F-actin, Trp-359 of PaExoY joins the hydrophobic cluster (Supplementary Figure 3C). We hypothesized that this movement might be detectable by a shift in intrinsic tryptophane fluorescence. Therefore, we measured tryptophane fluorescence of the PaExoY-F-actin complex and of the mixture of F-actin and PaExoY with a C-terminal deletion (PaExoY Δ C5) that abolish formation of the complex (an average of three measurements with standard deviation is shown below). Unfortunately, we did not see any shift of the peak maximum which is likely due to the strong background caused by 13 other tryptophanes in the PaExoY-F-actin complex. Nevertheless, we believe that our

cryo-EM and MD data convincingly demonstrate the conformational change in PaExoY upon formation of the PaExoY-F-actin complex.

There are grammatical errors throughout and the manuscript would benefit from a careful editing.

We thank the reviewer for pointing it out. We carefully proofread the manuscript and hope that there are no grammatical errors left.